# PVP1—The People's Ventilator Project: A fully open, low-cost, pressure-controlled ventilator research platform compatible with adult and pediatric uses

Julienne LaChance[1☯], Manuel Schottdorf[2☯], Tom J. Zajdel[3], Jonny L. Saunders[4], Sophie Dvali[5], Chase Marshall[6], Lorenzo Seirup[7], Ibrahim Sammour[8], Robert L. Chatburn[8], Daniel A. Notterman[9], Daniel J. Cohen[1]*

1 Department of Mechanical and Aerospace Engineering, Princeton University, Princeton, New Jersey, United States of America, 2 Princeton Neuroscience Institute, Princeton University, Princeton, New Jersey, United States of America, 3 Department of Electrical and Computer Engineering, Carnegie Mellon University, Pittsburgh, Pennsylvania, United States of America, 4 Department of Psychology and Institute of Neuroscience, University of Oregon, Eugene, Oregon, United States of America, 5 Department of Physics, Princeton University, Princeton, New Jersey, United States of America, 6 RailPod, Inc., Boston, Massachusetts, United States of America, 7 New York ISO, Rensselaer, New York, United States of America, 8 Department of Neonatology, Cleveland Clinic Lerner College of Medicine, Cleveland, Ohio, United States of America, 9 Department of Molecular Biology, Princeton University, Princeton, New Jersey, United States of America

☯ These authors contributed equally to this work.
* danielcohen@princeton.edu

**Data Availability Statement:** All relevant data are within the manuscript and its Supporting information files. All construction files and code are

## Abstract

Mechanical ventilators are safety-critical devices that help patients breathe, commonly found in hospital intensive care units (ICUs)—yet, the high costs and proprietary nature of commercial ventilators inhibit their use as an educational and research platform. We present a fully open ventilator device—The People's Ventilator: PVP1—with complete hardware and software documentation including detailed build instructions and a DIY cost of $1,700 USD. We validate PVP1 against both key performance criteria specified in the U.S. Food and Drug Administration's Emergency Use Authorization for Ventilators, and in a pediatric context against a state-of-the-art commercial ventilator. Notably, PVP1 performs well over a wide range of test conditions and performance stability is demonstrated for a minimum of 75,000 breath cycles over three days with an adult mechanical test lung. As an open project, PVP1 can enable future educational, academic, and clinical developments in the ventilator space.

## Introduction

The first mechanical ventilators date back to more than 150 years ago [1]. In the time since, they have undergone considerable design modifications; including, crucially, the transition from pure mechanical devices to the modern electronic machines in use today. Despite their

available from our documentation website (https://www.peoplesvent.org/en/latest/), and our public github repository (https://github.com/CohenLabPrinceton/pvp). The specific code and hardware underlying this publication are available on Zenodo at https://doi.org/10.5281/zenodo.5933282.

**Funding:** This work was supported by Princeton University which provided funding and facilities. In addition, JLS is supported by NSF Graduate Research Fellowship No. 1309047. TJZ was supported by the National Center for Advancing Translational Sciences (NCATS), a component of the National Institute of Health (NIH) under award number TL1TR003019. The sponsors and funders did not play any role in the study design, data collection and analysis, decision to publish, and preparation of the manuscript.

**Competing interests:** No competing interests.

commercial availability, very few platforms have been made open and fully transparent. Such a platform will enable the production of high-quality devices in virtually any laboratory, will further efforts in teaching and research/development, and may serve as development platform for a future medical tool [2–4].

In addition, over the past years, the global COVID-19 pandemic has highlighted the need for a low-cost, rapidly-deployable ventilator solution for the current and future pandemics. While safe and robust ventilation technology exists in the commercial sector, there exist a small number of suppliers who have been unable to meet the extreme demands for ventilators during a pandemic. Moreover, the specialized and proprietary equipment developed by medical device manufacturers can be prohibitively expensive and inaccessible in low-resource areas [4–9]. Ventilation as a technology is needed globally beyond pandemics for applications spanning neonatal intensive care, surgical anaesthesia, life support, and general respiratory treatments [10].

Finally, while the COVID-19 pandemic sparked a surge of interest in ventilation designs and some truly creative solutions, nearly all technologies put forth during this time have focused on evaluating performance with respect to adult guidelines. However, ventilation is of critical importance in pediatric medicine and it is valuable to consider developing a solution that is suitable for both adult and pediatric indications [10]. Hence there is a clear need for a broader range of solutions, both for research (e.g. to improve critical components [2]), clinical applications, and beyond [3, 7, 9, 11–15].

In response to these challenges, we present an open-source, rapid-deploy ventilator design with minimal reliance on specialized medical devices and manufacturing equipment. The People's Ventilator Project (PVP1) is a pressure-controlled and fully automatic mechanical ventilator that can be built for $1,700 by a single person in few days (cf. Fig 1). As a point of reference, the lower-end average market values of open ventilators such as the freely-released Puritan Bennett 560 [16] or the Mechanical Ventilator Milano [17] cost approximately $10,000. PVP1's parts were selected for widespread availability, and its modular software was designed to support component substitutions and extensions to new ventilation modes. Further, we have included comparisons here to commercial, pediatric-grade ventilators to emphasize the versatility of PVP1 and the goal of increasing global access to critical-care ventilation technology and making such technology available for teaching and research.

PVP1 is an automated ventilator that natively supports pressure-control ventilation (PCV), spontaneous inhalation monitoring ventilation (SIMV), and key alarms specified by regulatory agencies (e.g. high airway pressure, etc.). Pressure control was chosen over volume control because it is known to be safer [18] with respect to barotrauma risk, and SIMV was implemented because it increases the range of patients and conditions for which PVP1 can be used. Summarized in Fig 1, PVP1 operates as a computer-controlled, timed-cycle ventilator that requires only medical air and the patient-side respiratory tubing to be operated. To date, PVP1 has been set up three times by two different teams [2] and run continuously for over 300 hours with no alarms or failures noted (representative data from a physical test-lung shown in Fig 2).

## Materials and methods

PVP1 was designed as open source and transparent ventilation system. As such we have made all source code, electronics, bill-of-materials, complete CAD assembly, testing results, and relevant schematics open to the public (Fig 1). A certification is available from the Open Source Hardware Association (OSHWA; UID US002073). To aid discoverability, we have also generated an Open Know How Manifest (OKH-manifest), located in the hardware folder of PVP1's git repository. A snapshot of the current repository (v1.0, githash 0a22452) is available on

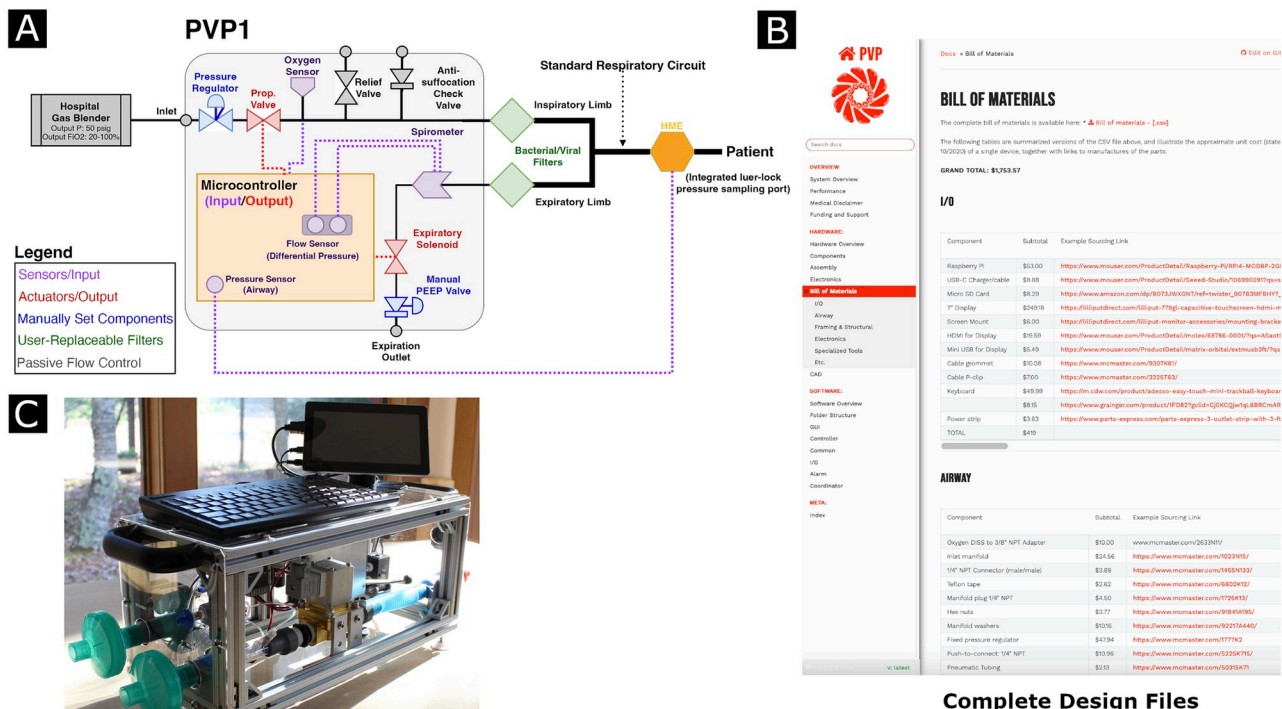

**Fig 1. A system overview of the People's Ventilator Project. A** Overview of respiratory circuit. **B** A snapshot of the online documentation in the form of a web-portal containing documentation and detailed build instructions. PVP1 Bill of Materials is highlighted. **C** The assembled device.

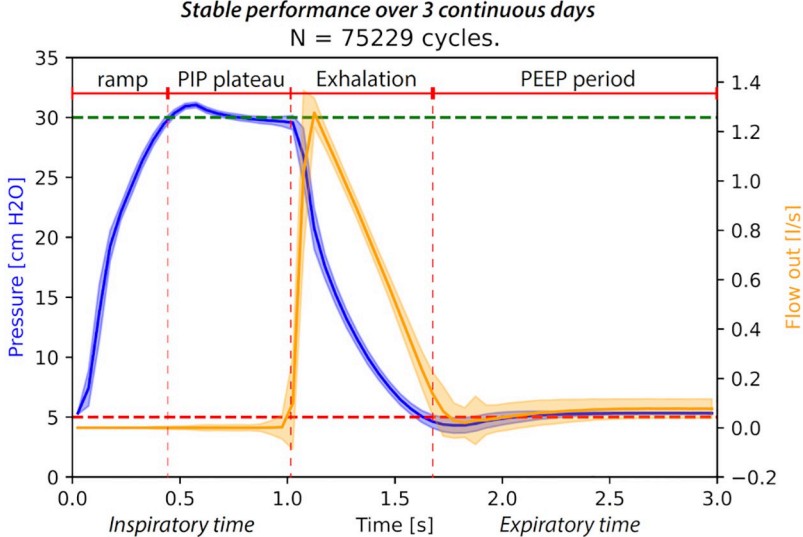

**Fig 2. Example of a PVP1 breath waveform.** Overlaid pressure control breath cycle waveforms for airway pressure and flow out over 70,000+ cycles, breathing into a Quicklung lung model. Test settings: compliance C = 20 mL/cmH$_2$O, airway resistance R = 20 cmH$_2$O/L/s, PIP = 30 cmH$_2$O, PEEP = 5 cmH$_2$O. Shaded areas indicate variability (standard-deviation) over the entire test period.

Zenodo [19]. For technical details, and complete documentation, we refer the reader to the Supporting Information, the online documentation and PVP1's git repository.

Briefly, PVP1 follows the FDA's (U.S. Food and Drug Administration) EUA (Emergency Use Authorization) key design criteria by automating the classic Manley ventilator design [5, 20]. A proportional–integral–derivative controller (PID) facilitates inflation of the lung to a set target pressure. This pressure is maintained for a set period of time. Then, air is allowed to escape passively through a second valve on the expiratory limb. The PVP1 schematic is shown in Fig 1A. The O2/air mixture is supplied to the system via a hospital gas blender. The patient breath cycle is actively controlled via a proportional valve on the inspiratory limb and a solenoid valve with mechanical PEEP (positive end-expiratory pressure) valve on the expiratory limb. The mechanical PEEP valve sets the positive pressure during expiration. This circuit is controlled by an embedded system (Raspberry Pi using the pigpio library [21]) supporting comprehensive monitoring of key alarm conditions, spontaneous breath detection and an intuitive touch-screen interface for clinician control. To use PVP1, the clinician programs a desired peak airway pressure (PIP), sets a manual PEEP valve to establish expiratory pressure, and sets a target respiratory rate and I:E ratio, the ratio of inspiratory to expiratory time. Convenient modifications to rise time and breath effort can be performed in real-time by the clinician. Core labeling specifications of PVP1 as required by the FDA EUA are presented in Table 1.

We feel that an open design should include justification of design decisions and be able to serve as a teaching and learning tool. As such, the Supporting Information here and online documentation aim to enable anyone to build PVP1, and to learn why and how key components were chosen. Based on independent validation from collaborators, following the guidelines will lead to a functional PVP1 in fewer than 24 work-hours ($\approx$ three eight-hour days). To further mitigate risk and expedite exploration and evaluation of the PVP1 platform, we

**Table 1. PVP1 specifications.**

| Parameter | Range | Comments: tested range, theoretical performance, notes |
|---|---|---|
| RR (BPM) | 10–40* | 12–20 tested based on ISO test tables [22]; *higher is feasible. |
| VTE (mL) | 100–500* | *Pressure controlled ventilation does not explicitly set VTE; we validated resultant VTE under PC as within the FDA EUA targets. |
| Flow Rate (L/min) | 0–100 L/m | Valve specifications; maximum needed for testing was 85 L/m. |
| PIP (cmH₂O) | 15–60 | Tested up to 35 cmH$_2$O during normal operation; safety hardware and alarms can support up to 60 cmH$_2$O as per FDA EUA guidelines. |
| PEEP (cmH₂O) | 5–25 | Validated PEEP range using approved commercial PEEP valves. |
| I:E Ratios | 1:1–1:3 | Tested based on ISO test tables [22]. |
| Available Ventilation Modes | PCV, SIMV | Modular system can be adapted for CPAP or non-invasive-modes at the software level. |
| Air Source | Hospital air | Rated for 50 psi, pre-blended oxygen/medical air mix. |
| Alarms | See Supplement | High/low airway pressure, hyper/hypoventilation, obstruction low FiO2, PEEP not met, Disconnect/high leakage, Technical Alarms. |
| Display variables | N/A | Airway pressure, expiratory flow, FiO2. Derived quantities: actual PIP, PEEP, estimated VTE. |
| Set variables | N/A | Target PIP, Target PEEP, Flow adjustment, Respiratory Rate, I:E ratio (or inspiratory time). |

Notes: Parameter ranges supported by PVP1. RR is respiratory rate, VTE is the Expiratory Tidal Volume, PIP is peak inspiratory pressure, PEEP is positive end-expiratory pressure, the I:E ratio is the ratio of inspiratory to expiratory phase of the breath cycle. PCV is pressure-control ventilation, and SIMV is spontaneous inhalation monitoring ventilation.

provide the ability to run a simulation of the PVP1 system on any computer. This simulation was also used for automated software tests.

# Results

## Core performance

In the following paragraphs, we will present a representative set of benchmarks and tests to demonstrate and validate PVP1 performance in key ventilatory processes. For clarity, we will focus on these results, followed by a more detailed discussion of how PVP1 was designed at the hardware and software level. More details and extended tests are provided in the S1 Appendix.

### Normal operating behavior

First, we evaluated the long-term stability and performance of PVP1 by performing standard pressure-controlled ventilation across more than 70, 000 contiguous cycles over a period of 3 days (Fig 2). All testing was performed using a high-grade test lung (Quicklung, Ingmar Medical) that offered the ability to tune compliance (C) and resistance (R) to meet FDA EUA test specifications (C = [5, 20, 50] mL/cmH$_2$O; R = [5, 20, 50] cmH$_2$O/L/s). Fig 2 shows pressure control performance for midpoint settings: C = 20 mL/cmH$_2$O, R = 20 cmH$_2$O/L/s, PIP = 30 cmH$_2$O, PEEP = 5 cmH$_2$O. PIP is reached within a 300 ms ramp period, then holds for the PIP plateau with minimal fluctuation of airway pressure for the remainder of the inspiratory cycle (blue). Once the expiratory valve opens, exhalation begins and expiratory flow is measured (orange) as the airway pressure drops to PEEP and remains there for the rest of the PEEP period.

Individual patient variation means that a one-size-fits all approach to pressure-controlled ventilation can have problems, and fine-tuning of key parameters such as the rise time (how quickly the ventilator reaches PIP) can allow more tailored ventilation. PVP1 supports such adjustment through a flow adjustment setting available to the clinician. This flow adjustment setting allows the user to increase the maximum flow rate during the ramp cycle to inflate lungs with higher compliance (briefly: this variable scales the proportional gain in the feedback control loop of the PID controller, similar to the mechanics of retinal contrast gain control [23]). The flow setting can be readily changed from the GUI and the control system immediately adapts to the user's input. An example of this flow adjustment is shown in Fig 3A for four breath cycles. While all cycles reach PIP, the latter two have a higher mean airway pressure, which may be more desirable under certain conditions than the lower mean airway pressure of the former two.

### Breath detection validation

A key feature of modern ventilators is to support spontaneous breaths should a non-anaesthetized patient try to breathe. Such patient-initiated breaths during the expiratory phase cause a sharp and transient drop in PEEP, and PVP1 can be set to detect these and trigger initiation of a new breath cycle. We tested this functionality by triggering numerous breaths out of phase, using a device (QuickTrigger, IngMar Medical, Pittsburgh, PA) to momentarily open the test lung during PEEP and simulate this transient drop of pressure (Fig 3B).

### Alarm response demonstration

Reliable and rapid alarm responses are a necessary feature of automated ventilators [24], and one of the most critical alarms ('high priority' in FDA EUA guidelines) for pressure control ventilation is the High-Airway-Pressure-Alarm (HAPA). According to peformance standards,

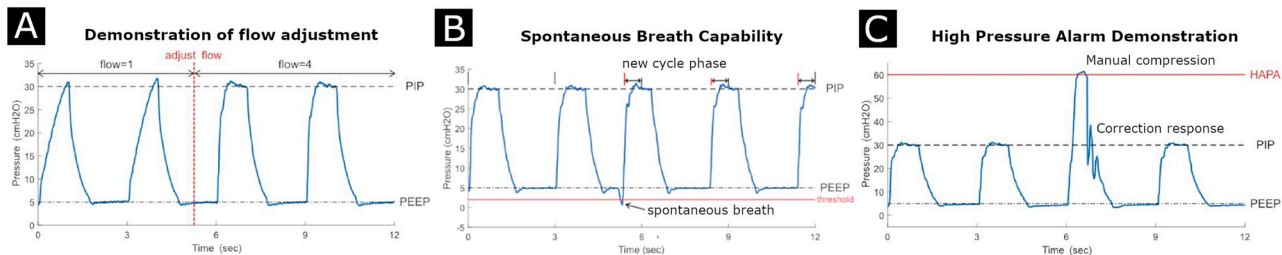

**Fig 3. Demonstration of waveform tuning and system capabilities. A** Demonstration of flow adjustment. If necessary, the operator can increase the flow setting through the system GUI to decrease the pressure ramp time. **B** Demonstration of a phase-shift if spontaneous breath is detected. **C** Demonstration of the response to a rapid high pressure transient. Test settings in all three cases: compliance C = 20 mL/cmH$_2$O, airway resistance R = 20 cmH$_2$O/L/s, PIP = 30 cmH$_2$O, PEEP = 5 cmH$_2$O.

the ventilator must detect and correct (within 2 breath cycles) abnormally high airway pressure. In PVP1, the HAPA alarm can detect and respond to elevated airway pressure within 500 ms, while also throwing a high priority visual and audible alarm (Fig 3C).

## PVP1 in a pediatric setting

PVP1 is generating breath waveforms using a simple and robust PID control scheme. In principle, this control scheme should allow reliably operation across physical lung parameters. To test the system's performance in the limiting case of an extremely small lung volume, we performed a number of experiments with a pediatric lung model (Ingmar QuickLung Jr.) and compared PVP1's performance to a Servo-I commercial mechanical ventilator. Recording the precise volume, pressure and flow waveforms allowed us to perform a comparison of both systems. The waveforms are shown in Fig 4. Notice that PVP1 rapidly delivers air at breath cycle onset (inset in Fig 4A) and then slows down. This suggests that the limitation is the PID-control-system, and not the physical hardware of the device, highlighting potential benefits of a more sophisticated control scheme. Overall, the pressure (Fig 4B) and flow (Fig 4C) waveforms

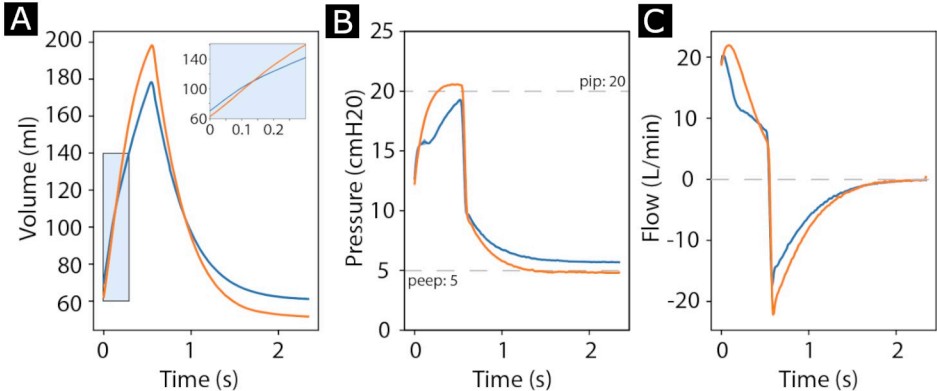

**Fig 4. PVP1 performance in a pediatric setting, compared to a Servo-I commercial ventilator. A** Volume of delivered air during a breath cycle. Blue is PVP1, Orange is Servo-I. The inset shows a magnified view of the first 0.3 s. Notice how the increase in volume is more rapid in PVP1 when compared to the commercial system. **B** Pressure waveforms produced by PVP1 and Servo-I. Dashed lines indicate target PEEP and PIP. **C** Flow in and out of the lung produced by both systems. For all three plots, we set the breath rate to 25 breaths per minute, PEEP to 5 cmH$_2$O, PIP to 20 cmH$_2$O, and inspiration time to 0.6 s. The pediatric lung settings were R = 25 cmH$_2$O/L/s and C = 10 ml/cmH$_2$O with an uncompensated residual capacity of 400 ml.

were similar, even though PVP1 deviated from target values slightly more than its commercial cousin (dashed lines in Fig 4B). The correlation coefficient between the waveforms of PVP1, and the professional ventilator were $r = 0.990$ for pressure, $r = 0.999$ for volume, and $r = 0.979$ for flow, reflecting the high similarity.

To assess the deviations of PVP1 from the set values, and compare the results with the commercial systems on long time scales, we collected several hundred breath cycles, and compared averaged statistics and standard deviation. In this second experiment, we obtained peak pressure values of $(21.5 \pm 0.4)$ cmH$_2$O, a PEEP of $(5.7 \pm 0.1)$ cmH$_2$O and inspired tidal volume of $(128 \pm 3)$ ml of air, for the same settings as in Fig 4. These averaged metrics were comparable to Servo-I which delivered: Peak pressure $(20.6 \pm 0.01)$ cmH$_2$O, PEEP $(4.79 \pm 0.01)$ cmH$_2$O, Inspired Tidal Volume $(146.9 \pm 0.1)$ ml. Across experiments, PVP1 deviated from the target pressure by $\approx 1$ cmH$_2$O, and delivered an inspired tidal volume $\approx 13\%$ below that of Servo-I in a challenging pediatric setting.

## Discussion

PVP1 is a flexible, stable, and open platform for pressure controlled ventilation with a total cost of 1300 USD for low-volume production. PVP1 is open source, featuring detailed documentation, automated software tests, and modular design. It offers anyone a state-of-the-art platform for exploring mechanical ventilation. For documentation and source code, we refer the reader to the Supporting Information, and the online documentation and the git repository. In the remainder of this section, we will discuss the history, key areas for improvement and performance notes worth bearing in mind for those considering PVP1 for different use-cases.

### Overall performance assessment

PVP1 has demonstrated sustained operation over at least 70,000 continuous cycles without failure while maintaining stable ventilatory performance using a default test condition from the EUA test table. PVP1 reaches PIP, and required VTE for key EUA tests. The greatest deviation from target PIP, by 9%, was observed under challenging conditions, when ventilating with very high airway resistances (50 cmH$_2$O/L/s; see S1 Appendix). These are uncommon patient conditions and PVP1 performs significantly better in all other test cases. Future improvements to better compensate for challenging patient conditions could involve updates to the proportional valve to allow for finer control over flow, as well as more advanced control schemes to better modulate overshoot [2].

### Design considerations in a pandemic

In response to the COVID-19 pandemic, the scientific community developed numerous exciting ventilator projects, many of which leverage earlier designs developed to combat prior respiratory pandemics such as SARS and H1N1 [5]. When designing PVP1, we sought to learn from the challenges and limitations noted in these prior studies while aiming for an open and transparent design. There are two key ventilator designs that received early FDA Emergency Use Authorizations—The University of Minnesota Ventilator [25] and the Mechanical Ventilator Milano [17]. To handle production and FDA EUA approval, both projects eventually shifted manufacturer-of-record status to major companies—Boston Scientific and Elemaster, respectively. Other academic projects such as the Vent4Us/PezGlobo ventilator (Stanford / University of Utah / University of Delaware) [26] have merged over time and also incorporated a variety of commercial backers. Still other projects such as the MIT E-Vent bag-valve ventilator [27], the RapidVent system [11] and the Portsmouth Ventilator [12] have remained

open, but with a more limited scope. A more comprehensive discussion of numerous ventilator projects can be found in [5], where it is highlighted that the most successful projects have necessarily become less open due to constraints from industrial partners. Hence, a key goal with PVP1 was to describe and demonstrate a fully functional ventilation platform that both highlights how effective a minimal design can be and provides a fully open platform for the broader community to leverage.

PVP1 was designed to align with the constraints and demands of a pandemic such as COVID-19, and special care was taken to specify reliable, commercial, off-the-shelf components outside of the traditional ventilator or scientific supply chains. Most components are available from general hardware suppliers and the chosen parts listed here did not experience supply chain disruptions due to COVID-19 during the period of development. The internal layout and chassis design are also sufficiently modular and simplistic to allow PVP1 to be adapted to a given clinical context without altering function. Modular and well-documented code facilitates simple adaptation of the system to different hardware. Finally, we hope PVP1 can either directly or indirectly improve access to ventilators even beyond the COVID-19 pandemic, while also offering a reliable and open research platform for further ventilator development.

## Design process and critical iterations

PVP1 went through multiple iterations of its software and hardware. To be as open and transparent as possible, the following paragraphs discuss the challenges of this process, as well as describe the evolution of the design. With regards to hardware decisions, some brief context on the landscape of ventilator methodology will help explain key choices made in PVP1's design. First, the Ambu-bag (a manual resuscitator) squeezers: Ambu-bags are very low-cost devices which require an attendant to squeeze a bag to force air into the lungs of a patient. These may have valves which help maintain PEEP, and external devices designed to continuously squish the bags [15, 25]. However, it is uncommon to leave a patient on these bags for long periods of time: notably, COVID-19 patients in critical condition need ventilator support for 2–3 weeks, and these devices are hard to precisely and digitally control. Second, CPAP/ BiPAP machines: CPAP (continuous positive airway pressure) machines are used by people with sleep apnea and are intended for use with patients who need some assistance breathing but can breathe on their own. The patient does not need to be sedated to use a CPAP, as they just involve a mask on the face. However, while CPAP machines are excellent instruments for ventilation [8], they cannot sustain high pressures like more invasive ventilators can, and thereby cannot sustain patients in more critical conditions. Critical COVID-19 patients' lungs become stiffer and ultimately an invasive ventilator is required. Additionally, there is a concern that CPAP machines may pose risk of spreading the airborne viruses. BiPAP machines are slightly more advanced than CPAP machines (multiple pressures can be set) but associated pros/cons are similar. Finally, invasive ventilation remains both the standard-of-care for many medical procedure, and cost prohibitive in many localities. Hence, while the design of PVP1 ensured compatibility with severe respiratory conditions such as COVID-19, we also sought to ensure that PVP1 would be valuable beyond the current pandemic. We therefore decided to build a ventilator that allowed ventilation support for several weeks to maximize the versatility of PVP1. Overall, PVP1 hardware went through three major design changes that we refer to as Mk1, Mk2 and Mk3, illustrated in Fig 5. CAD files for all three versions are provided in the online materials and details are covered in the following paragraphs.

Mk1 was the mechanically simplest version, realizing a pressure limited time cycled continuous flow ventilator. Such a device connects the patient to a continuously flowing air supply.

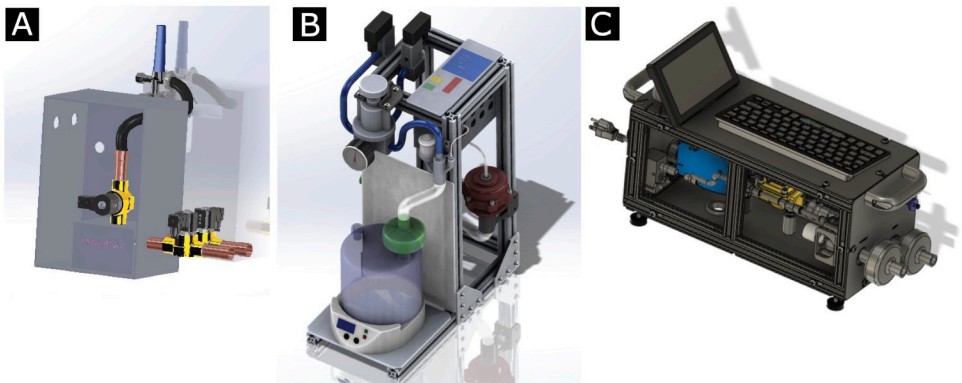

**Fig 5. The two earlier versions of PVP1, and its current design. A** Mk1 of PVP1 was a simple pressure limited time-cycled continuous flow ventilator. **B** Mk2 was a significant step up in complexity. Its upright design featured an air humidifier, and numerous valves and sensors. At this point we had also committed to pressure controlled ventilation. **C** Mk3 is the up-to-date version of PVP1, described in this article, featuring a screen and keyboard, and tailored to the supply chain limitations. To aid mechanical stability, we also transitioned to a flat design.

The only controlled element is a single expiratory valve. If this valve is closed, the lung inflates. With the valve open, the patient can exhale. PEEP pressure is maintained by the continuous airflow. While this system is very simple and robust, it lacks precise control of ventilation parameters, and is relatively wasteful regarding medical oxygen, a commodity that became a key limiting factor in many localities during the COVID-19 pandemic. Mk2 addressed these concerns, realizing a full pressure controlled ventilator.

Mk2 used wall O2/medical air from the hospital wall, passing it through a gas blender, a humidifier, and then controlling the flow (using a series of devices, including a pressure regulator, expiratory valves, etc) to the endotracheal tube of the patient. Mk2 allowed to set upper pressure values (PIP), lower pressure values (PEEP), and other relevant parameters (breath rate, inspiratory time, etc.). The sensors of this version were specified to specific operating conditions: high flow, low pressure, and sometimes high humidity. However, supply chain limitations constrained numerous choices in the early design. For example, we initially planned to use a servo-proportional valve for fine control of inspiratory flow (RCV-075, Enfield Technologies, Trumbull, CT), but sourcing such valves operating at low pressures (0–50 cmH$_2$O) and sufficiently high flow (over 120 LPM) was challenging due their being needed for existing commercial ventilator design. Therefore, few manufacturers were willing or able to provide us with valves, due to existing contracts with ventilator manufacturers. Moreover, such specialized valves are often costly and thus antithetical to the goal of a broadly accessible ventilator design. While more available, generic industrial proportional valves are typically not designed to run at clinically-relevant pressures or flow rates. Likewise, the expiratory valve selection was constrained to those valves which would open even with zero pressure differential because the pressures exhibited inside the human respiratory tract are far lower than in a typical industrial process. Again, these factors heavily constrained what was possible during the COVID-19 pandemic and what would be possible in a resource limited setting moving forward. The Mk3 design emerged specifically to combat the valving constraints encountered in Mk2, namely: high cost, lack of availability of advanced valves, difficulty sourcing valves that could operate in the physiological range. In addition, we incorporated a number of critical user-interface and functionality upgrades in Mk3. First, Mk3 was rotated down to a table-top or rack-based configuration to improve stability. A screen was also incorporated to make the unit entirely self-

sufficient (Mk2 had assumed a clinic-provided display). We froze structural elements of the design here as well, including the use of imperial standard fittings. Additionally, at this stage in PVP1's design process, it became untenable to source additional flow sensors, and we made the decision to run PVP1 entirely with a single flow sensor at the inspiratory side of the circuit —a cost and computational saving measure that precluded the use of volume-controlled ventilation. As for the valves, we made two key decisions that largely removed us from supply chain limitations. First, we identified an industrial current-controlled proportional valve (PVQ31–5G-23–01N) with a sufficient performance envelope and high input pressure to support human ventilation and obviate the need for more expensive and less accessible low-pressure-high-flow medical valves (see Methods). Next, we decided to couple a low-cost, accessible solenoid valve (ON-OFF) to a standard, fully-mechanical commercial PEEP valve to regulate expiratory with minimal financial and computational cost. Noting that even commercial PEEP valve supply can be disrupted, we also provide a prototype for a 3D-printable membrane-based PEEP-valve with the online materials. Finally, the new form factor of Mk3 allowed us to significantly reduce tubing length while increasing tubing stiffness. This reduced compliance proved essential to enabling PVP1 to achieve therapeutic performance during pediatric ventilation simulation. We were particularly determined to ensure pediatric compatibility given the lack of pediatric testing or consideration with any existing open ventilators, thereby again increasing the utility of PVP1 within and beyond a pandemic.

## PVP1: An open source project during a pandemic

PVP1 aims to follow the best practices for open projects [28]: (i) transparent and public communication on the GitHub repository, (ii) standardized and automated tests of individual modules and the full system with >99.5% coverage using Travis CI [29], (iii) merger of pull requests only after passed tests and independent code review, (iv) high-level hardware and build documentation, and API-level documentation generated from docstrings and (v) PVP1 made accessible via pip and the Python Package Index which allows anyone to easily install, run, and experiment with the software.

Designing a device like PVP1 during the COVID-19 pandemic provided us with unique challenges that we wish to briefly summarize here. Historically, this project started around April 2020, early in the pandemic, from a small group of core developers. As PVP1 matured, more expertise was required, but the strict social distancing and remote-work requirement made it challenging to build a community, i.e. a broader group of developers personally committed to developing this device without pay or other compensation. To address this, we employed various methods. First, we obtained external help by experts in crowdsourcing projects, on citizen science, and science communication. To this end, we leveraged internal connections within the university to reach out to such experts, who amplified our needs on social media, such as Twitter. Second, we have published multiple university-level press releases to access and recruit people outside of the limited scope of our personal social media presence. In addition, we set up a Slack channel, a Discord server, and a public git repository on GitHub as a platform for interested people to engage. In addition, to aid discoverability, we have also generated a standardised Open Know How Manifest (OKH-manifest).

We found collaboration in small sub-groups of around three people to be most effective. These groups communicated in a self-organized way, and were concerned with conceptual parts of the ventilator (such as software/control, electronics, and mechanics/hardware). During the pandemic, we remained in continuous contact, and the entire team all met at least once per week to monitor overall progress and identify bottlenecks. Over the last two years, we have grown to a core team of about 10 developers, plus another 10 who contributed temporarily.

At the current scale with around 20 contributors, GitHub proved viable to maintain and curate this work. However, our aim is to develop a growing and active community to continuously improve this product. For added transparency, and better support for large groups of contributors, a Wiki would be a viable and desirable addition to the PVP1 ecosystem.

## The case for open source medical technology

Open source medical technology [3, 6, 9] can improve the capability and access to medical technology as a whole in several ways: (1) enabling faster device innovation with lower costs [3, 4]; (2) increasing economic value, with associated public benefits, compared to traditional proprietary development [30]; (3) facilitating external review and inspection by avoiding black-box hardware and software designs, and (4) providing a benchmark for innovation towards next-generation technology such as smart ventilators [31]. Finally, the open source approach can make these problems more accessible to academic researchers, thereby greatly expanding the ability to train students in approaching such problems through hands-on open-ended pedagogy [32] as well as encouraging unconventional approaches [2, 7]. While many pandemic ventilator projects began as open-source initiatives, these often transitioned to a closed format due to the strong structural and regulatory incentives to enter into industrial partnerships. With PVP1, we provide a completely open build guide and software platform for a functional, pressure-controlled ventilator designed for FDA Emergency Use Authorization standards with viability in both adult and pediatric settings.

The world is moving towards more open technology. Other projects of the medical instrumentation universe [3, 4, 9] that were recently published include an open peristaltic pump build around an arduino controller [33], a syringe pump build using a Raspberry Pi [34], and a low-cost positive airway pressure ventilation system, working with water-columns for pressure control [10]. These examples show that particularly in low-resource settings, open medical [9], but also scientific instrumentation (e.g. [35, 36]) are becoming a reality. In particular, some of these items can constitute valuable components of a revised version of PVP1, such as the pressure sensor developed by Goertzen and colleagues [37] or the monitoring system developed by the Princeton Open Ventilation Monitor Collaboration [38].

## Challenges of open hardware

There is rough consensus in the open source software community around some basic development best practices like version control and automated testing. Open hardware has relatively few analogous best practices. One challenge is the lack of open formats that support versioned designs, instead most rely on proprietary CAD programs with opaque binary formats. For PVP1's software, we were able to track almost 1,000 commits in the git repository, numerous issues in the issue tracker, and 73 descriptive merge requests. This is much harder to do in hardware.

We have briefly summarized important design changes in the earlier paragraphs, but systematic documentation and version control of hardware is an ongoing challenge in many fields. For example, one author had to implement a system for hardware knowledge organization from scratch in their Autopilot wiki [35]. Documenting the many hardware components, CAD files, and usage guides used with the system required self-hosting an instance of semantic mediawiki and populating it with hundreds of properties, forms, and templates. This system is still only a partial replacement for the version control and dependency specification tools available for software, though it is an improvement over traditional hardware design repositories like thingiverse, open neuroscience, and others that are typically composed of static documents organized with a uni-dimensional "tag" field. We describe the autopilot wiki here as an

illustration of the challenge of exhaustive hardware documentation that constrains our ability to fully reconstruct PVP's design history, as well as an open space in tooling that we have attempted to fill in subsequent projects.

We attempted to follow the best practices for open source hardware that do exist. PVP1 is OSHWA-compliant (OSHWA UID US002073), and we provide a standardised Open Know How Manifest in the hardware folder of our git repository. Our hardware documentation includes a high-level overview, full solidworks models including two prior designs, descriptions of the system components and reasoning behind their selection, complete assembly instructions with accompanying pictures of each step, CAD files for the custom electronics and 3D printed components, and a bill of materials with purchase links. Since the hardware documentation is automatically built on readthedocs.org from source files included in the software repository, we also welcome questions, clarifications, and contributions using issues and pull requests.

## Future hardware development goals

PVP1 is released as a minimal implementation of a safe, invasive ventilator capable of Pressure Controlled Ventilation with spontaneous breath detection. There are, of course, many ways that the software and hardware design can be improved. Indeed its continual improvement is the point: we have developed and documented the system such that it is not a static design, but can be modified and improved as a general-purpose ventilation platform. We welcome programmers and users to submit issues to discuss bugs and needed developments, and submit their own improvements via pull requests, or in their own branches. PVP is intended to be a continually, communally developed project. We specifically invite others to contribute to the project, and consider this reviewed report as a solid foundation for future developments.

A useful upgrade would be to incorporate an inspiratory flow sensor which would further open the possibility of Volume Controlled Ventilation and allow for the use of inspiratory flow for improved PSV. However, as PVP1 is inherently modular, both in terms of hardware and software, these features can genuinely be added both to the open code base and to the assembly with minimal complication.

Another useful future development goal is a version of PVP1 with metric parts. While imperial parts are readily available from sources like McMaster-Carr in the United States, and certain countries in the Commonwealth of Nations, it is important to note that such parts may not be readily available in many countries. The same holds for certain standardized medical parts, where we again followed US standards. We hope that people with similar limitations will find our designs transparent enough to overcome supply limitations.

## Future software development goals

Modifications made purely at the software level (e.g. a firmware upgrade) would allow PVP1 to additionally support complete Pressure Supported Ventilation (PSV, for spontaneously breathing patients) as well as Non-Invasive Ventilation (such as Continuous Positive Airway Pressure, CPAP). We did not implement these at the present time as they were considered luxury-features in a pandemic ventilator.

In addition, automatic ventilator data collection can eliminate delays, improve charting efficiency, and reduce errors caused by manual entry of data [39]. Standards are specified in ISO/IEEE 11073, describing communication between medical, and health care devices with external computer systems [40]. PVP1 is easy to integrate into existing software. Its data logger already supports the export of all raw data into hdf5 and standard data formats (MatLab's .mat and comma-separated-values .csv). A future version could provide a RS-232 or other interface

for digital output, and automatically insert data into SQL-tables to facilitate integration into a patient's file. Similarly, since the xml-rpc inter-process communication module operates over a network socket, it is straightforward to allow centralized control or monitoring of PVP1 in hospital settings [39].

## Missing steps towards a medical device

While anyone can build PVP1 in around 24 work hours, we explicitly emphasize that it is not a legally licensed medical device (it currently lacks US-FDA regulatory approval or that of any other regulatory body). Anyone producing PVP1 for clinical use would need to take on the legal responsibility of Manufacturer-of-Record (MoR) and seek appropriate regulatory approval. Moving PVP1 forward thus primarily requires formal regulatory approval beginning with certifications of achieving key international performance and safety standards as discussed below.

While we built PVP1 to address the most critical performance targets and safety alarm integrations specified in the US-FDA Emergency Use Authorization (EUA) for ventilators, a manufacturer-of-record would need to take on liability and formalize these results. More specifically, several important international standards exist that regulate medical devices. Particularly relevant are the international standards ISO 13485, ISO 14971 and IEC 62304 [41–43]. These set standards for clinically used hard- and software, more specifically regarding requirements for quality (ISO 13485 [41]) and risk (ISO 14971 [42]) management system. IEC 62304 specifies international standards for life cycle requirements regarding software within medical devices [43]. In this report, we have focused on FDA EUA guidance (which is largely harmonised with international standards), but only partially followed these international norms given the unprecedented nature of the COVID-19 pandemic. It should also be noted that international standardization is an ongoing challenge in a multilateral work. The Global Harmonization Task Force (GHTF) and its successor, the International Medical Device Regulators Forum (IMDRF) have made important progress in this regard, but much remains to be done.

## Tuning

PVP1 is built around sensitive pressure-sensors that can be subject to drifting when exposed to extreme environments (such as air travel). Re-adjustment and tuning can easily be performed with air of known pressure (available in nearly all medical settings), and copying the voltage readings of the sensors to the program code. In general, this will be necessary in regular intervals, or after travel. Failure to calibrate the sensors properly can lead to deviations on the order of a few $cmH_2O$ and the recommended calibration routine would become part of the formal product labeling and appropriate use documentation.

## Customization

PVP1's completely open design and modular code base makes customization trivial. For example, the GUI can be adjusted to fit various screen sizes, resolutions or color schemas. It can also be easily manipulated by removing, or adding panels and information. In fact, we have often operated PVP1 from a simple LCD external monitor, and even remotely via a Secure Shell connection (ssh). In the latter case, a webcam is useful to independently monitor the experimental setup. This enables the potential for telemedicine in a future crisis.

The internal code can also be customized easily. For example, it is straight forward to swap-out the controller with a more complex piece of code [2]. The hardware abstraction layer

allows easy calibration and tuning of the sensors, while also allowing for complete replacement of a software- and hardware-module if a particular sensor becomes unobtainable.

## Performance in a pediatric setting

Waveforms produced by PVP1 were similar to a commercial ventilator. Importantly, initial rise time is very rapid, suggesting that PVP1's hardware is indeed viable across very different patient settings. The relatively slow inflation during inhalation might be related to conservatively chosen PID constant, more specifically a too large integration (I) term. A large integration term is useful to avoid ringing in the limit of high flow rates, i.e. to rapidly inflate large, adult lungs. In the pediatric setting with small lung volume, this coefficient can decreased. Tailoring the PID coefficients to the pediatric settings will likely improve PVP1's performance considerably.

## Supporting information

**S1 Appendix. Further tests, and validation data of PVP1.** The appendix contains the complete set of EUA ISO standard tests [22], elaborates on the design, and provides calibration and validation data.
(PDF)

## Acknowledgments

We would like to thank Grant Wallace, Zhenyu Song, Moritz Kütt and Philippe Bourrianne for very valuable discussions and technical support during the development of PVP1, and Amy Sterling, the EyeWire team, and Sebastian Seung for their help and guidance in building a PVP1 community. In addition, we would like to thank Elad Hazan and Daniel Suo with the Google AI team at Princeton for testing the quality of PVP1's build instructions. We would also like to acknowledge the contribution of the open science community as a whole, by providing guidelines, standards and tools.

**Disclaimer**: PVP1 is not a regulated or clinically validated medical device. We have not yet performed testing for safety or efficacy on living organisms. All material described herein should be used at your own risk and does not represent a medical recommendation. PVP1 is currently recommended only for research purposes.

This document is not connected to, endorsed by, or representative of the view of Princeton University. Neither the authors nor Princeton University assume any liability or responsibility for any consequences, damages, or loss caused or alleged to be caused directly or indirectly for any action or inaction taken based on or made in reliance on the information or material discussed herein.

PVP1 is under continuous development and the information here may not be up to date, nor is any guarantee made as such. Neither the authors nor Princeton University are liable for any damage or loss related to the accuracy, completeness or timeliness of any information describe or linked to from this website.

## Author Contributions

**Conceptualization:** Julienne LaChance, Manuel Schottdorf, Chase Marshall, Lorenzo Seirup, Daniel A. Notterman, Daniel J. Cohen.

**Data curation:** Julienne LaChance, Manuel Schottdorf, Tom J. Zajdel, Jonny L. Saunders, Daniel J. Cohen.

**Formal analysis:** Manuel Schottdorf, Tom J. Zajdel.

**Funding acquisition:** Daniel J. Cohen.

**Investigation:** Manuel Schottdorf, Tom J. Zajdel, Sophie Dvali, Ibrahim Sammour.

**Methodology:** Julienne LaChance, Manuel Schottdorf, Tom J. Zajdel, Jonny L. Saunders, Chase Marshall, Lorenzo Seirup, Daniel A. Notterman, Daniel J. Cohen.

**Project administration:** Manuel Schottdorf, Daniel J. Cohen.

**Resources:** Julienne LaChance, Manuel Schottdorf, Tom J. Zajdel, Jonny L. Saunders, Chase Marshall, Lorenzo Seirup.

**Software:** Julienne LaChance, Manuel Schottdorf, Jonny L. Saunders, Lorenzo Seirup.

**Supervision:** Daniel J. Cohen.

**Validation:** Manuel Schottdorf, Tom J. Zajdel, Sophie Dvali, Ibrahim Sammour, Robert L. Chatburn.

**Visualization:** Julienne LaChance, Manuel Schottdorf, Tom J. Zajdel, Jonny L. Saunders, Sophie Dvali.

**Writing – original draft:** Julienne LaChance, Manuel Schottdorf, Tom J. Zajdel, Jonny L. Saunders, Daniel J. Cohen.

**Writing – review & editing:** Julienne LaChance, Manuel Schottdorf, Tom J. Zajdel, Jonny L. Saunders, Daniel J. Cohen.

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
