## [Decision Letter · Decision Letter 0]

16 Dec 2021

PONE-D-21-35089PVP1 - The People's Ventilator Project: A fully open, low-cost, pressure-controlled ventilator research platform compatible with adult and pediatric usesPLOS ONE

Dear Dr. Schottdorf,

Thank you for submitting your manuscript to PLOS ONE. After careful consideration, we feel that this manuscript has potentially great value to the community, but in its former state, the documentation is not sufficient for current Open Hardware Standards. We think this could be solved by addressing the reviewers comments. Therefore, we invite you to submit a revised version of the manuscript that addresses the points raised during the review process. Please submit your revised manuscript by Jan 29 2022 11:59PM. If you will need more time than this to complete your revisions, please reply to this message or contact the journal office at plosone@plos.org. Please include the following items when submitting your revised manuscript:A rebuttal letter that responds to each point raised by the academic editor and reviewer(s). You should upload this letter as a separate file labeled 'Response to Reviewers'.A marked-up copy of your manuscript that highlights changes made to the original version. You should upload this as a separate file labeled 'Revised Manuscript with Track Changes'.An unmarked version of your revised paper without tracked changes. You should upload this as a separate file labeled 'Manuscript'.

We look forward to receiving your revised manuscript.

Kind regards,

Andre Maia Chagas, PhD

Academic Editor

PLOS ONE

Journal Requirements:

This work was supported by Princeton University which provided funding and facilities.

JLS is supported by NSF Graduate Research Fellowship No. 1309047. We would also

like to thank Grant Wallace, Zhenyu Song, Moritz K¨utt and Philippe Bourrianne for

very valuable discussions and technical support. In addition, we would like to thank

Elad Hazan and Daniel Suo with the Google AI team at Princeton for testing the

quality of our build instructions. We would also like to acknowledge the contribution of

the open science community as a whole, by providing guidelines, standards and tools.

We note that you have provided funding information. However, funding information should not appear in the Acknowledgments section or other areas of your manuscript. We will only publish funding information present in the Funding Statement section of the online submission form. 

This work was supported by Princeton University which provided funding and facilities. In addition, JLS is supported by NSF Graduate Research Fellowship No. 1309047. The sponsors and funders did not play any role in the study design, data collection and analysis, decision to publish, and preparation of the manuscript. 

Reviewers' comments:

Reviewer's Responses to Questions

**Comments to the Author**

1. Is the manuscript technically sound, and do the data support the conclusions?

Reviewer #1: Yes

Reviewer #2: Partly

2. Has the statistical analysis been performed appropriately and rigorously? 

Reviewer #1: N/A

Reviewer #2: I Don't Know

3. Have the authors made all data underlying the findings in their manuscript fully available?

Reviewer #1: Yes

Reviewer #2: Yes

4. Is the manuscript presented in an intelligible fashion and written in standard English?

Reviewer #1: Yes

Reviewer #2: Yes

5. Review Comments to the Author

Reviewer #1: This paper presents the People's Ventilator Project (PVP1), a fully open-source and low cost ventilator which can serve as a research platform. While it has not been medically certified yet, it shows some promising results with artificial lungs in both adult and paediatric cases.

Overall, the paper is well written and provides an important addition to the literature. They authors make a strong case for the need for open source ventilators, while also highlighting that numerous other open source ventilators had to shift away from being open source due to external (primarily commercial and regulatory) pressures. The latter points to a need in societal changes to accommodate for open projects.

The authors can find my comments within the attached the PDF file. In addition to those, I'd like to mention that it is important that the images and tables move so that they are in a location that fits the flow of the paper. Right now, they really disrupt the reading experience. As a rule of thumb, it is good for figures and tables to follow after they have been introduced in the main body text. In conclusion, I am happy to recommend this paper for publication subject to the authors considering making the recommended minor changes.

Reviewer #2: The manuscript presents and open hardware ventilator for educational and research purposes. The authors appear to have taken considerable care to ensure that the ventilator is also appropriate to pediatrics. Some thought has been taken to how the design may be deployed as a medical device but this aspect is some way off. The authors correctly identify this and they should be commended for being explicit and clear about the current status of their project.

For context of this review my expertise is on open hardware design and transferring open hardware designs from within academia into distributed production in the global south. This review does not consider the ventilator performance itself. This aspect should be assessed by a reviewer with specific expertise.

The manuscript regularly describes the project as "fully open", boasting that a "complete" web-portal is "optimized" for sharing the entire design, and that complete design decisions have been provided. The software has indeed been developed using open source best practices. A github repository with almost 1,000 commits, and with numerous issues in the issue tracker, and with 73 descriptive merge requests. From this I am confident I could review and understand the software design. All software documentation is versioned in the same repository and rendered. However, this is far from the case for the hardware. The assembly instructions are a Google document, the rendered bill of materials is truncated on the website making it impossible to read. The original CAD files needed to continue the design are provided. However, they all dumped haphazardly into a Google drive folder titled "Julie Magic Air Machine", complete with names such as "I_AM_AN_IDIOT_ADAPTER_15M_TO_22F.SLDPRT". There appears to have been no attempt to document the design history, nor to curate and archive the design somewhere more stable than a Google Drive.

I suggest that at the very least the team organise and curate the design files into a directory structure complete with README files in place to help someone navigate. Any design history that is available could be captured in CHANGELOG files, perhaps with older versions in sub directories. These files should then be archived somewhere such as Zenodo so that this manuscript can link to an unchanging identifier.

To clarify, I pick out the "idiot adapter" not to say that when we make mistakes they should be hidden, however without a structured history the context of what this means is lost. Mistakes are also recorded in the software history, but they are removed from the current version and identified clearly with descriptive commit messages.

Considering design decisions, what is provided is in the supplementary information. A few pages of top down design information do little to mitigate the vast quantities of design information that has been lost by not following the same excellent open practices that were followed for the software design. Of course the processes for collaborative open hardware design are far less established than they are for software. The authors would do better to tone down the self-congratulation on how fully open they are and address head on the challenges they had in working openly on the hardware design. Comparing and contrasting these challenges to the standard practices that they mention for the software design would be both useful for future reference and would be more honest. The author should look to the literature for other commentary or examples on the technical challenges of open hardware design, both generally and specifically in the case of open design of ventilators during the pandemic.

Moving on..

The authors state that no specialist knowledge is required to reproduce the design. It should be clarified that this would not be true in the case of a medical device. It is worth explicitly clarifying that anyone looking to take the design to market would need to fully understand the device and take responsibility for it as a legal manufacturer. Of course this is why the design decisions mentioned above are so critical. The authors would do well to link the sharing of design decisions to medical device manufacturing in the context of international standards for medical device design such as ISO 13485 and ISO 14971. Due to the heavy software component IEC62304 should also be considered. I understand that the ventilator is not at the stage of a medical device, but putting the work needed to transition it to one in the context of the internationally harmonised standards is more useful than only considering only the FDA guidance (which is harmonised with international standards). The authors may find that documentation for the Global Harmonisation Task Force will help them put the work a more international context.

Furthermore it is a shame that the ventilator has been designed using USA-only threads and fittings. My experience working in Africa is that when we have tried to replicate American designs that use USA-specific fixing, we need to significantly modify the design before it can be implemented. USA threads and fittings are available in limited capacity in Europe, but they require highly specialised suppliers. The rest of the world has standardised on metric, having spent years working in the USA I know that metric components are readily available from sources like McMaster-Carr. I suggest that the manuscript should temper its claims to have selected readily available components, or be specific that the components are only readily available in one single country.

In conclusion the paper describes a ventilator designed during the pandemic. This ventilator has done more than many by ensuring that the orignally design files are available. However, reality of the open hardware design does not meet the rose-tinted description in the manuscript, which largely implies (incorrectly) that best practices used for software were also followed for the ventilator. The manuscript and design are also highly USA focused. With curation and archiving of the hardware design, and significant tempering of the language, tone, and descriptions in the manuscript I believe this work should be published in PLOS ONE.

6. PLOS authors have the option to publish the peer review history of their article (what does this mean?). If published, this will include your full peer review and any attached files.

Reviewer #1: **Yes: **Rafaella Antoniou

Reviewer #2: **Yes: **Julian Stirling

---

## [Author Response · Author response to Decision Letter 0]

24 Feb 2022

We have submitted, as pdf files, the following items:

- A letter to the editor.

- A rebuttal letter that responds to each point raised by the reviewers, named 'Response to Reviewers'.

- A marked-up copy of our manuscript that highlights changes made to the original version, in blue. This file is labeled 'Revised Manuscript with Track Changes'.

- An unmarked version of our revised paper, without tracked changes, labeled 'Manuscript'.

---

## [Decision Letter · Decision Letter 1]

29 Mar 2022

PVP1 - The People's Ventilator Project: A fully open, low-cost, pressure-controlled ventilator research platform compatible with adult and pediatric uses

PONE-D-21-35089R1

Dear Dr. Schottdorf,

We’re pleased to inform you that your manuscript has been judged scientifically suitable for publication and will be formally accepted for publication once** the minor revision comments from the reviewers are addressed and all outstanding technical requirements have been fufilled**.

Kind regards,

Andre Maia Chagas

Academic Editor

PLOS ONE

Additional Editor Comments (optional):

Reviewers' comments:

Reviewer's Responses to Questions

**Comments to the Author**

1. If the authors have adequately addressed your comments raised in a previous round of review and you feel that this manuscript is now acceptable for publication, you may indicate that here to bypass the “Comments to the Author” section, enter your conflict of interest statement in the “Confidential to Editor” section, and submit your "Accept" recommendation.

Reviewer #2: All comments have been addressed

Reviewer #3: (No Response)

2. Is the manuscript technically sound, and do the data support the conclusions?

Reviewer #2: Yes

Reviewer #3: Yes

3. Has the statistical analysis been performed appropriately and rigorously? 

Reviewer #2: Yes

Reviewer #3: Yes

4. Have the authors made all data underlying the findings in their manuscript fully available?

Reviewer #2: Yes

Reviewer #3: Yes

5. Is the manuscript presented in an intelligible fashion and written in standard English?

Reviewer #2: Yes

Reviewer #3: Yes

6. Review Comments to the Author

Reviewer #2: I would like to thank the authors for making significant changes to the Discussion section manuscript. The section on the design process and critical iterations is interesting and well written. I also think that their new "Challenges of open hardware" section is very fair and balanced. I am very interested in their AutoPilot Wiki for hardware knowledge organisation, I shall dig into it in more detail over the coming weeks.

Whist reading the manuscript I notices the following minor formatting issues:

* One of the LaTeX quotes around "tag" is backwards in the "Challenges of open hardware" section

* The ISO standards are mentioned by name but have not been added to the reference list.

I apologise for the tone of my original review, some deep seated frustrations were contributed to a somewhat hostile tone. I thank the authors for their diligent work in updating the manuscript.

Reviewer #3: First, I would like to thank the authors for the time and efforts they put into developing and documenting this PVP1 project.

The authors highlight a clear lack of accessibility to open device and documentation for respiratory systems, and the need for more open designs in this field.

The manuscript clearly presents a pressure-controlled ventilator system. The documentation detailed in this manuscript and on the associated repositories follows open-source guidelines and it is quite pleasant to observe that this project documentation went through the OSHWA certification process before its submission for publication. In addition, the OKH manifest is a great addition though I fail finding this project on openknowhow.org. Finally, the modularity of the system described makes it reachable and upgradable to all.

The development history of the project is very enjoyable to read to allow the reader to get a proper insight on the developmental approaches, successes, and failure along with the reasoning that led to this reliable and modular system.

It is also quite appreciable to find a medical disclaimer on the manuscript and on all repositories.

I do not have any major comment regarding the last version of this manuscript and would recommend it for publication in PLOS.

I have however a few minor comments. I hope the authors will see them as my attempt to contribute to their project with suggestions to make it both more robust and user friendly.

The authors claim that their device could be beneficial for research and teaching, however I fail to understand the research application for their device. As I am completely ignorant of this field of study maybe a quick example would help the lambda reader to fully appreciate the potential of this project.

Is there any reason for an inspiratory flow sensor not to be installed in the current version as the authors themselves admit it would be a useful addition with minimal implementation complication?

Also, I can only encourage the authors, as an European, to make a metric version for global reproduction without modification from the user part to accommodate their own standard units. Even if I agree with the authors that their design is clear enough to be modified. Still this little effort will save time and trouble to so many. Otherwise, the rest of the building parts are available worldwide; it is unfortunately not a systematic concern to most OSH developers, and it is great to see the authors operating in this direction.

For the software development part, I admit my ignorance on hospital record databases. As I might not be the only one, a quick contextual sentence would give credit to the authors for their desire to extend their I/O interface. Also, is that an educational/research need?

Would it be possible to just quickly explain what is meant by paediatric settings (inflation during inhalation time I presume) so the reader could understand why a longer integration in your PID is relevant to these patients.

The assembly instructions are clearly annotated and easy to follow. It is appreciable that you share your PCB supplier for electronics neophytes that may wonder how to replicate your custom PCBs, however your gerber files are not easy to find, and would deserve to be highlighted on your pvp.org page as you do for your CAD files both in pvp/asset and direct download like the CAD parts.

A similar comment could be made for the software part. Personally, I appreciate the details of your documentation, but for future upgrade I would recommend adding an installation manual for people not use to work with Python libraries, raspberry Pi or any kind of electronics/programming. This is a trivial comment that does not concern the quality of the manuscript work, rather an observation that might suit the spirit of this project to be accessible to the greatest number.

While discussing trivial points, I have noticed a couple of typos:

-In the introduction page2 line 9: “there exists a small number”

-In the introduction page2 line 10: “during the/the last pandemic”

-In the results page6 line 9: consider fixing the syntax error: “[…], the ventilator must detect abnormally high airway pressure must be detected and corrected within 2 breath cycles”

- the acronyms EUA, VTE and PID were not defined, and PEEP was not defined at its first occurrence (first seen at table1 legend)

Overall, this is an impressive OSH project with a high-quality documentation available on multiple platform that still hold the promise to future upgrade.

Thank you to the authors and editor for sharing this manuscript to my review

7. PLOS authors have the option to publish the peer review history of their article (what does this mean?). If published, this will include your full peer review and any attached files.

Reviewer #2: **Yes: **Dr Julian Stirling

Reviewer #3: **Yes: **Maxime Zimmermann

---

## [Editor Report · Acceptance letter]

18 Apr 2022

PONE-D-21-35089R1 

PVP1–The People’s Ventilator Project: A fully open, low-cost, pressure-controlled ventilator research platform compatible with adult and pediatric uses 

Dear Dr. Schottdorf:

I'm pleased to inform you that your manuscript has been deemed suitable for publication in PLOS ONE. Congratulations! Your manuscript is now with our production department. 

Kind regards, 

on behalf of

Dr. Andre Maia Chagas 

Academic Editor

PLOS ONE